# Acetone–Ether–Water Mouse Model of Persistent Itch Fully Resolves Without Latent Pruritic or Cross-Modality Priming

**DOI:** 10.3390/dermatopathology12010005

**Published:** 2025-02-11

**Authors:** Zachary K. Ford, Adam J. Kirry, Steve Davidson

**Affiliations:** 1Neuroscience Graduate Program, College of Medicine, University of Cincinnati, Cincinnati, OH 45267, USA; fordzk@mail.uc.edu; 2Department of Anesthesiology, College of Medicine, University of Cincinnati, Cincinnati, OH 45267, USA; kirryam@ucmail.uc.edu; 3NYU Pain Research Center, Department of Molecular Pathobiology, College of Dentistry, New York, NY 10010, USA

**Keywords:** itch, dry skin, priming, pruritis, AEW, epidermis

## Abstract

Hyperalgesic priming is a model of the transition from acute to chronic pain. Whether a similar mechanism exists for “pruritic priming” of itch is unknown. Here, we tested the hypothesis that itchy skin in a commonly used mouse model of dry skin pruritus develops latent sensitization after resolution. Acetone–ether–water (AEW) treatment induced a dry and itchy skin condition in the mouse cheek that elicited site-directed scratching behavior. After cessation of treatment and the complete resolution of AEW-induced scratching, histaminergic and non-histaminergic pruritogens were administered to the cheek to test for altered site-directed scratching and wiping behavior. Each pruritogen was also tested following the resolution of carrageenan-induced nociceptor hypersensitivity to test for cross-modality priming. Peak AEW-induced scratching occurred 24 h after the final day of treatment, and 5 days were required for scratching levels to return to baseline. Likewise, epidermal thickening was the greatest on the final treatment day and completely returned to baseline after 5 days. After the resolution of itchy cheek skin, acute histamine- and non-histamine-evoked scratching and wiping behaviors were unchanged, nor were scratching and wiping behaviors to acute pruritogens altered after the resolution of carrageenan-induced hypersensitivity. The results indicate that persistent itch due to dry skin likely resolves completely, without producing a latent primed response to subsequent pruritic stimuli. We conclude that the mechanisms regulating hyperalgesic priming are likely distinct from pruritic signaling in the dry and itchy skin model.

## 1. Introduction

Chronic pruritus is an aversive somatosensory experience commonly associated with inflammatory skin disease, end-organ disorder, and neuropathy that significantly and negatively impacts the quality of social, emotional, and physical life [1,2]. Itch from many pruritic diseases, including atopic dermatitis, xerosis, and psoriasis, responds poorly to H1 antihistamines, yet treatments for non-histaminergic itch are only just emerging and remain nonspecific [3,4]. Pruritic skin disease often presents with skin barrier dysfunction, loss of epidermal water content, and epidermal thickening [5,6,7,8,9]. These symptoms can be modeled in rodents by treating the skin repeatedly with acetone, ether, and water (AEW) which disrupts barrier function, produces dry skin, and induces site-directed scratching [10,11,12,13].

Several lines of evidence indicate that AEW-evoked scratching is a useful animal model of non-histaminergic itch and, therefore, may be a clinically relevant tool for the investigation of non-histaminergic pruritic diseases that respond poorly to H1 antihistamines. Genetic depletion of histamine-producing mast cells in the skin does not alter AEW-evoked scratching, and acute behavioral hypersensitivity to pruritic stimuli (hyperknesis) after AEW occurs only with non-histaminergic compounds but not with histamine [13,14,15]. Likewise, AEW treatment sensitizes the physiological responses of primary sensory neurons to the non-histaminergic pruritogen chloroquine but not to histamine [11]. Moreover, AEW treatment produces increased gene expression of the chloroquine-sensitive receptor MrgprA3 and increases the epidermal fiber density of MrgprA3-positive fibers, suggesting an important role of the histamine-independent MrgprA3 receptor in dry skin itch [12]. MrgprA3-positive neurons are a unique subset of pruriceptive chemo-nociceptors that express non-peptidergic markers, including IB4, as well as peptidergic markers [16].

Hyperalgesic priming is a phenomenon in which a noxious stimulus delivered to the same site on the body after recovery from an earlier injury or insult results in heightened and longer-lasting sensitivity [17,18]. Hyperalgesic priming is used to model the transition from acute to chronic pain and has elucidated potential targets to suppress this transition. Several of the cellular mechanisms underlying hyperalgesic priming are restricted to the subset of IB4-positive, non-peptidergic sensory fibers [19,20,21]. Because MrgprA3 fibers express IB4 and have a recognized role in itch, we hypothesized that a mechanism analogous to hyperalgesic priming could exist for pruritus, that is, “pruritic priming.” Therefore, we established the time course for recovery from AEW-induced dry skin itch and then tested whether pruritogen-evoked scratching behavior could be primed. We also tested whether cross-modality priming could be evoked whereby dry skin itch or inflammatory pain could prime the other modality after a full recovery.

## 2. Materials and Methods

### 2.1. Animals

All experiments adhered to National Institutes of Health guidelines and were approved by the University of Cincinnati Institutional Animal Care and Use Committee. Adult male C57Bl/6 mice (Jackson Labs) between 8 and 12 weeks old were used for all experiments. The mice were given ad libitum access to both food and water and were housed in a 12 h/12 h light/dark cycle. Hyperalgesic priming is sexually dimorphic, and female sex hormones protect against the development of priming rodents [22]. Therefore, to assess the first model of pruritic priming, only males were used in this study.

### 2.2. AEW Pruritus Model

Dry, itchy skin was generated using the acetone–ether–water (AEW) model, as done previously [11]. Cheek skin was used because mice scratch the cheek with the hind limb when pruritogens are applied but wipe the cheek with the forelimb when painful agents are applied, thus providing a means to differentiate between itch and pain behaviors [23]. The cheek skin was shaved unilaterally with grooming clippers, and a 1:1 mixture of acetone (Fischer Scientific) and diethyl ether (Sigma) was applied for 30 s. This was followed by a 30 s application of water using a soaked gauze or a cotton-tipped swab. Control animals were treated with water only. This procedure occurred twice each day (morning and evening) for 5.5 days. The cheek skin was transected from separate cohorts of mice at four time points during AEW treatment: Baseline, day 6, day 7, and day 12. The tissue was immersed in Zamboni’s fixative for 4 h, rinsed with PBS, and then placed in 30% sucrose. To assess skin anatomy, hematoxylin and eosin (H&E) staining was performed on cheek tissue fixed in the same manner, as before, but embedded in paraffin and cut to 4 µM sections.

### 2.3. Behavioral Assessments

*Scratching/Wiping:* In order to assess scratching behavior, mice were placed individually in Plexiglas containers (10.5 cm × 10.5 cm × 15.5 cm) separated by opaque cards to block visual contact between animals. Their behavior was recorded on video, and wiping and scratching bouts were counted offline. Due to the scaling of cheek skin, blinding was not possible. Mirrors were placed on the back and sides of the chamber to obtain a full view of the mice from all directions. A bout of scratching was defined as any number of individual scratching events separated by a pause. Only site-directed scratches to the ipsilateral cheek were counted. Wiping behavior was defined as a rostrally directed movement of the ipsilateral forelimb across the cheek starting from the ear [23].

### 2.4. Pruritic Priming Model

Baseline spontaneous scratching and wiping behaviors were recorded for 1 h prior to starting treatment. After 5.5 days of twice-per-day treatments of AEW or water alone, spontaneous site-directed scratching and wiping behaviors were recorded for 1 h each day. After recovery, the mice were injected intradermally with a 30-gauge insulin syringe in the previously treated cheek with a pruritogen: chloroquine (CQ, 100 µg/10 µL; Sigma, St. Louis, MO, USA), histamine (50 µg/10 µL; Sigma), or β-alanine (100 mM; Sigma). Scratching and wiping behaviors were then recorded. A single pruritogen was tested three times within the same cohort, each test separated from the previous one by 4 days. For test cross-modality priming in the cheek, the mice were given an intradermal injection of 1% carrageenan in 10 µL (*w*/*v*; Sigma) or a vehicle injection of 0.9% saline [24]. A minimum period of 5 days was set to allow for resolution. Next, the mice were tested with pruritogens, as mentioned before.

### 2.5. Data Analysis

All statistical analyses were performed using GraphPad Prism 9. Comparisons between the time points of the AEW time course, CQ priming, histamine priming, cross-modality CQ, and cross-modality histamine were carried out using two-way ANOVA, followed by multiple comparisons, as appropriate. The total number of scratching bouts was evaluated using an unpaired t-test. Comparison of the epidermal thickness across time points was performed by one-way ANOVA with Tukey’s multiple comparisons. Significance was defined as *p* < 0.05. Data were presented as the mean ± SEM.

## 3. Results

### 3.1. Time Course of Behavioral and Anatomical Recovery After Cessation of AEW Treatment

The onset of scratching behavior during AEW treatment is well documented [10,11,25]; however, recovery from dry skin and scratching behavior after cessation of treatment is not reported. Consistent with previous studies, AEW treatment induced significant scratching within 6 days of starting treatment (Figure 1A). Surprisingly, peak scratching occurred one full day after cessation of treatment. Thereafter, scratching steadily decreased toward baseline over the next 4–5 days. Epidermal thickening occurred with AEW treatment, reaching the maximum value in tandem with scratching behavior 1 day after the end of treatment (day 7). Epidermal thickness returned to baseline with a time course over the next 4–5 days, in line with the reduction in scratching behavior (Figure 1B–C). These results indicate that both pruritus and epidermal anatomy return to baseline together by 5 days after AEW treatment.

### 3.2. AEW-Treated Skin Does Not Produce Latent Priming of the Response to Acute Pruritogens

We tested the hypothesis that after recovery from itchy, dry skin, a latent primed state to pruritic sensitivity is induced. This would complement what has been shown after recovery from inflammation and the development of latent hyperalgesic priming [18,26,27,28]. Latent sensitization may occur in primary afferent fiber types responsive to certain pruritic compounds. Therefore, we examined three distinct pruritogens that operate through separate principal receptors: histamine (H1R), chloroquine (MrgprA3), and β-alanine (MrgprD). Five days after AEW-induced scratching returned to baseline (day 16), an intradermal injection of each pruritogen was administered to the cheek. Prior AEW treatment produced no changes in the scratching frequency or duration from histamine, CQ, or β-alanine (Figure 2A–C).

### 3.3. Previous Painful Stimulus Did Not Affect Future Scratching Behavior

We next determined whether a prior noxious stimulus followed later by itch-producing stimuli could produce a priming effect of the scratch response. Mice were injected intradermally in the cheek skin with the hyperalgesic priming stimulus carrageenan (1% *w*/*v*). After 5 days of recovery, the mice were given an intradermal injection of a pruritogen, or after 7 days, AEW treatment (Figure 3A–D). No difference in scratching behavior was seen.

## 4. Discussion

Dry skin pruritus is associated with several chronic skin diseases, such as psoriasis and atopic dermatitis [9]. In this study, the time course for the resolution of scratching and the reversal of epidermal thickening was determined for the commonly applied AEW mouse model of dry skin pruritus. After cessation of AEW treatment, spontaneous scratching behavior and skin thickness recovered and returned to baseline at 4–6 days. These data were needed to design an experiment to test the hypothesis that itch, like pain, can induce latent sensitization after recovery. If true, such latent sensitization, or “pruritic priming”, could be further investigated in future studies as a mechanism for the poorly understood but clinically important transition from acute to chronic pruritus. The cheek skin model was leveraged to differentiate itch-evoked hind limb scratching from ambiguous, nocifensive murine behaviors that occur on the nape of the neck or other parts of the body [23].

The AEW model was specifically chosen to test pruritic priming because, in addition to establishing that it fully recovers, the neurons responsible for signaling dry skin itch share many of the same characteristic markers with the neurons responsible for hyperalgesic priming. Hyperalgesic priming occurs when nociceptors that have seemingly recovered from an earlier insult subsequently produce heightened and longer-lasting sensitivity after a second insult [17,18]. Molecular mechanisms that are integral for hyperalgesic priming have been localized to the subset of IB4-positive, non-peptidergic sensory nociceptive neurons and entail intracellular signaling involving protein kinase C (PKC) [19,20,21]. IB4-positive, non-peptidergic neurons have, therefore, been linked importantly to the transition from acute to persistent pain.

Dry skin itch has also been linked to IB4-positive, non-peptidergic nociceptive neurons. After AEW treatment, non-peptidergic fiber innervation increases in the epidermis [11,29]. Furthermore, AEW-evoked scratching requires the cation channel TRPA1 [30,31], and TRPA1 is expressed primarily in IB4-positive, non-peptidergic rodent sensory neurons [32]. TRPA1 receives intracellular signals from the chloroquine-activated pruritic receptor MrgprA3, also localized to IB4-positive neurons [16,33]. Ablation of Mrgprs suppresses AEW-induced scratching, and AEW treatment produces an increase in chloroquine-sensitive sensory neurons [11,12,34]. Taken together, these data indicate that the sensory neurons responsible for AEW pruritus share defining characteristics of the neurons responsible for hyperalgesic priming. Therefore, we hypothesized that a mechanism analogous to hyperalgesic priming could exist for dry skin itch.

AEW-evoked dry skin itch is considered to be non-histaminergic because (1) H1 antihistamines are generally ineffective for dry skin itch [35] and (2) AEW scratching develops normally in mast-cell-deficient mice [10]. Although both histamine and non-histaminergic pruritogens activate different receptors, both histamine and non-histaminergic receptors (e.g., Mrgprs) can be found on the same neuron and may engage overlapping intracellular signaling pathways [16,36]. All the pruritic receptors examined in this study, H1R, MrgprA3, and MrgprD, are transmembrane G-protein-coupled receptors with an extracellular ligand-binding domain for their respective pruritogens: histamine, chloroquine, and β-alanine. The histamine receptor H1R, when examined in heterologous cell systems, canonically signals through a Gq- and PLC/PKC-dependent pathway [37]. Downstream, H1R can engage the ion channels TRPV1 and may also activate TRPA1, which are important for the signaling of itch [38]. MrgprA3, a pruritic receptor activated by chloroquine, has been shown to engage TRPA1 and Piezo2 ion channels and also involves phospholipase-C/PKC [39,40,41]. The β-alanine-sensitive MrgprD subset of primary afferent fibers, which are also IB4 positive and implicated in non-histaminergic itch, similarly have been shown to engage PLC signaling [42,43,44]. Therefore, due to the overlap in intracellular signaling pathways engaged by these distinct pruritic receptors, we hypothesized that cross-sensitization after dry skin itch could occur and, therefore, tested histaminergic and non-histaminergic pruritogens after the recovery of AEW-induced dry skin. Interestingly, nearly all of the characterization of the intracellular signaling pathways of pruritic receptors has been performed in heterologous systems or mouse dorsal root ganglia. Therefore, the human counterparts for these receptors, MrgprX receptors and human H1R, remain poorly understood in native human sensory neurons, where their functions are clinically important.

Each pruritogen, histamine, chloroquine, and β-alanine, was effective in eliciting site-directed scratching; however, we found no evidence of a primed response for any of the compounds after AEW treatment. We next asked whether the scratching response to these pruritogens might be primed instead by pretreatment in the cheek with the classic hyperalgesic priming mediator carrageenan [18]. Similarly, no evidence for cross-modal pruritic priming after recovery from carrageenan with histamine, chloroquine, β-alanine, or dry skin was seen. These data suggest that pruritic priming from dry skin as a mechanistic complement to hyperalgesic priming may not occur. Interestingly, peripheral sensitization of pruriceptors, as defined by lowered pruritic activation thresholds or a greater response magnitude, does occur and produces alloknesis and hyperknesis, or “itchy skin”, in human and animal models [45,46,47]. Clinically relevant pruritic diseases, such as atopic dermatitis, share mechanisms with peripheral pain; however, there are also important distinctions [48]. Thus, while pruriceptors can be sensitized and produce behavioral hypersensitivity in other models, the long-term latent sensitization underlying hyperalgesic priming is not matched with dry skin itch.

A key strength of this study is that it is the first investigation of the novel concept of pruritic priming as a complement to the prior and ongoing well-designed studies that have characterized hyperalgesic priming [18]. A rational approach was taken to choose an appropriate model to test pruritic priming that mimics the key features of hyperalgesic priming, that is, recovery to baseline within several days and activation of non-peptidergic nociceptors and PKC signaling. Several limitations of the study should also be recognized. We tested the AEW model using only a 5.5-day treatment protocol. It is possible a longer or more robust dry skin pruritus may yield a behavioral priming effect. To equate scratching magnitude with pain behavior magnitude is not straightforward; nevertheless, we selected an itch model that resolved behaviorally in 4–6 days, similar to the injection of carrageenan or nerve growth factor often used in hyperalgesic priming protocols. It should be noted that in addition to non-peptidergic fiber hyperinnervation, AEW increases peptidergic innervation [29,49] and that MrgrpA3-positive neurons express peptidergic markers as well [16], suggesting that the specific population of itch nociceptors might uniquely express both non-peptidergic and peptidergic markers, making them a distinct subpopulation from the pure non-peptidergic nociceptors responsible for some types of hyperalgesic priming. Other pruritic dermatitis models, such as the vitamin D3 analog [50] and imiquimod [51], may yet yield pruritic priming. Additionally, we used only a single dose of pruritic compounds to test for priming near the midpoint of their dose response; however, it is possible lower doses may have revealed a priming effect. Finally, our study tested only male mice, and it remains an open question as to whether female mice could exhibit a different result.

In summary, our results indicate that the resolution of AEW-induced scratching subsides concurrently with dynamic changes in the epidermal thickness. A prior bout of dry-skin-induced pruritus is not sufficient to produce a latent primed state. Additionally, prior inflammation does not produce cross-modality sensitivity to histaminergic or non-histaminergic itch. We conclude that if latent pruritic priming does exist, it may operate with mechanisms distinct from hyperalgesic priming.

## Figures and Tables

**Figure 1 dermatopathology-12-00005-f001:**
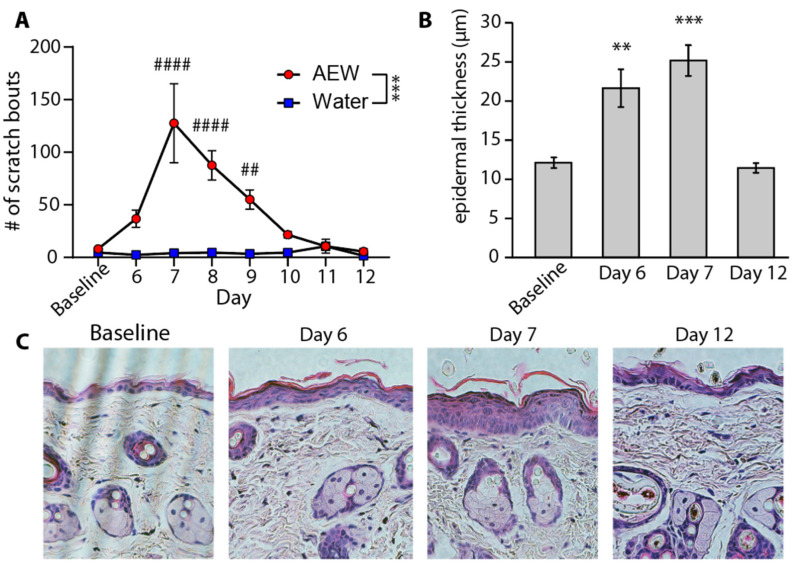
Time course of recovery from AEW-evoked scratching and epidermal thickness. (**A**) Quantification of the mean number of scratch bouts during 1 h of observation at each time point: (*n* = 16/group days 0–7, *n* = 8/group days 8–12) two-way ANOVA, *** *p* < 0.001; Holm–Sidak post-tests, ## *p* < 0.01, #### *p* < 0.0001 (asterisk refers to ANOVA; hashtag refers to Holm-Sidak post-test). (**B**) Epidermal layer thickness at BL and on D6, D7, and D12 (*n* = 4 each time point) over the AEW time course (one-way ANOVA, ** *p* < 0.001, *** *p* < 0.001). (**C**) H&E staining shows hyperplasia of the epidermis and dissociation of the stratum corneum.

**Figure 2 dermatopathology-12-00005-f002:**
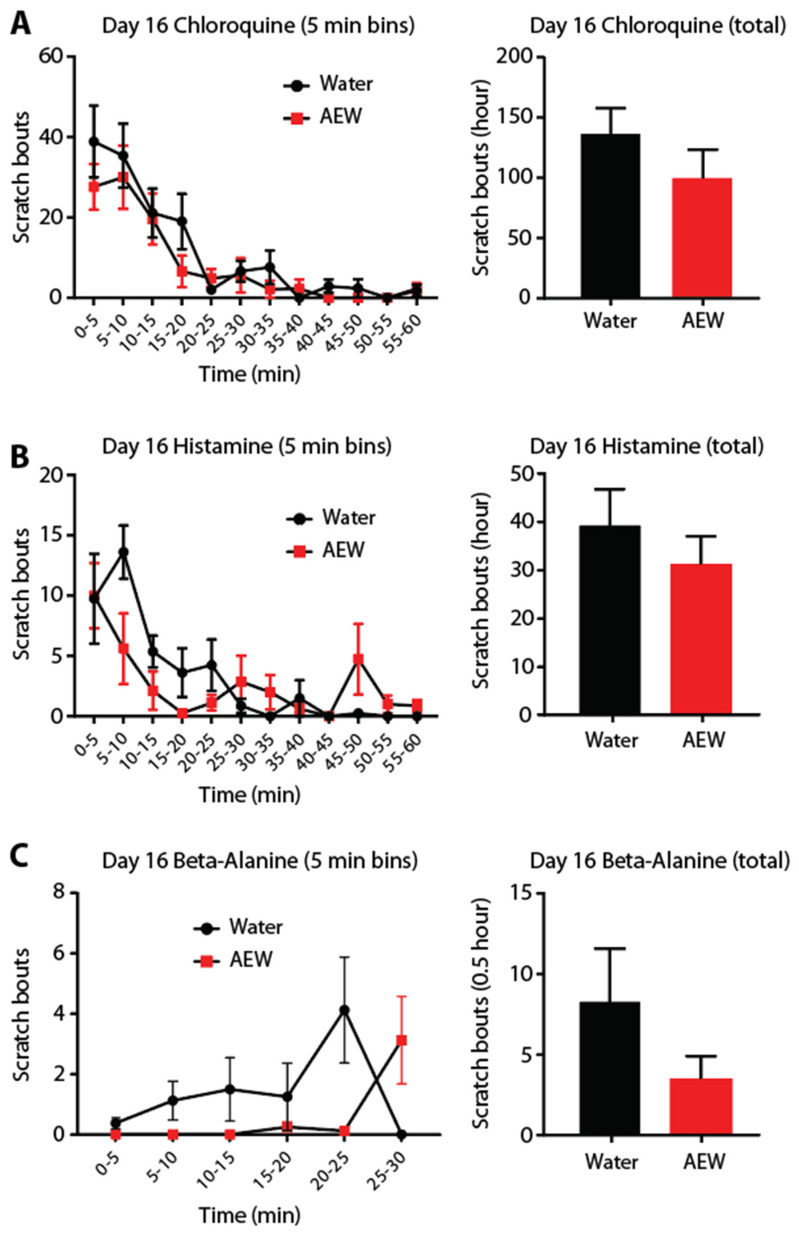
Acute pruritogens following AEW treatment do not alter scratching behavior. Five–minute binned and total number of scratching bouts after chloroquine, histamine, or β-alanine injection at different time points following recovery from AEW treatment (**A**–**C**). No significant difference was found in the amount of scratching between the water and AEW groups (*n* = 8 each group).

**Figure 3 dermatopathology-12-00005-f003:**
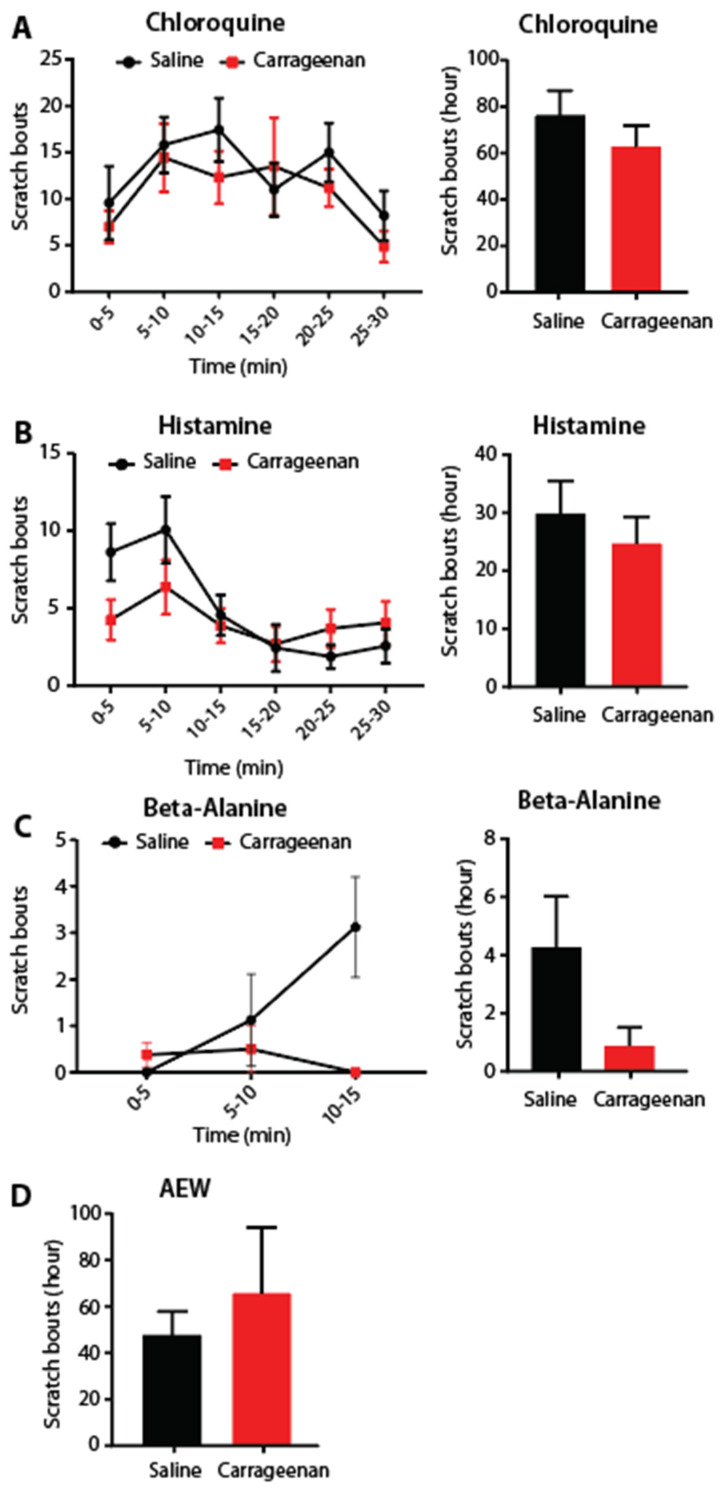
Prior inflammation did not alter subsequent pruritogen-induced scratching. (**A**,**B**) Five–minute binned and total number of scratching bouts after (**A**) chloroquine, (**B**) histamine, (**C**) β-alanine, and (**D**) AEW treatment following a prior saline or carrageenan injection to the cheek. No significant difference was found in the amount of scratching between the saline and carrageenan groups (CQ and Hist, *n* = 16 each group; β-ala and AEW, *n* = 8 each group).

## Data Availability

Datasets are available on request from the authors.

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
