# Peer review of "Acetone–Ether–Water Mouse Model of Persistent Itch Fully Resolves Without Latent Pruritic or Cross-Modality Priming"

_dermatopathology, 2025, doi:10.3390/dermatopathology12010005_

Round 1
Reviewer 1 Report
Comments and Suggestions for Authors
1. The authors need to provide a clearer explanation of why distinguishing between histaminergic and non-histaminergic itch mechanisms in the AEW model is significant.
2. I think the manuscript could benefit from a more detailed discussion of the pathways involved in MrgprA3 and MrgprD signaling.
3. I think the discussion on pruritic priming could include more context about its potential clinical relevance.
4. I suggest adding a dedicated section to explicitly discuss the strengths and limitations of the study.
5. I suggest testing varying doses of histamine, chloroquine, and β-alanine to examine whether a dose-dependent response exists, providing further insights into the pathways involved.
6. The authors need to perform a simple RT-PCR analysis of MrgprA3 expression in skin samples from AEW-treated mice could validate the molecular findings at the tissue level.
7. I suggest adding a histological analysis at intermediate recovery time points (e.g., Days 3 and 4) to provide a more detailed view of the dynamics of epidermal repair.
Author Response
Review 1
- The authors need to provide a clearer explanation of why distinguishing between histaminergic and non-histaminergic itch mechanisms in the AEW model is significant.
We thank the reviewer and have added text to the introduction (p. 3 lines 12-13) to address the significance of histamine and non-histaminergic itch in the AEW model.
“Several lines of evidence indicate that AEW-evoked scratching is a useful animal model of non-histaminergic itch and therefore may be a clinically relevant tool for the investigation of non-histaminergic pruritic diseases that respond poorly to antihistamines.”
Additionally, we have added and clarified text in the discussion (p. 6 lines 23-26) explaining how pruritic priming after AEW could lead to greater histamine sensitivity to better explain the rationale for the histamine-evoked scratching assessment after the recovery from AEW.
“…histamine and non-histaminergic itch receptors can be found on the same neurons [16; 35]. Therefore, we tested both histaminergic and non-histaminergic pruritogens after the recovery of AEW reasoning that either class of pruritic compound might exhibit neuronal pruritic priming.”
- I think the manuscript could benefit from a more detailed discussion of the pathways involved in MrgprA3 and MrgprD signaling.
We agree and have added additional material to the discussion on the topic of MrgprA3, MrgprD, and H1R signaling pathways on pp. 6-7, lines 28-43, 1-2. We have also included several additional relevant citations on the subject (36-43).
“Each of the pruritic receptors examined in this study, H1R, MrgprA3, and MrgprD, are transmembrane G-protein coupled receptors with an extracellular ligand binding domain for their respective pruritogens: histamine, chloroquine, and β-alanine. The histamine receptor H1R, when examined in heterologous cell systems, canonically signals through a Gq and PLC/PKC dependent pathway [36]. Downstream, H1R can engage the ion channels TRPV1, and may also activate TRPA1, which are important for the signaling of itch [37]. MrgprA3, a pruritic receptor activated by chloroquine, has been shown to engage TRPA1 and Piezo2 ion channels and also involved phospholipase-C/PKC [38; 39; 40]. The β-alanine sensitive MrgprD subset of primary afferent fibers, which are also IB4-postive and implicated in non-histaminergic itch, similarly have been shown to engage PLC signaling [41; 42; 43]. Therefore, due to the overlap in intracellular signaling pathways engaged by these distinct pruritic receptors, we hypothesized that cross-sensitization after dry skin itch could occur and therefore tested histaminergic and non-histaminergic pruritogens after the recovery of AEW dry skin. Interestingly, nearly all of the characterization of the intracellular signaling pathways of pruritic receptors has been performed in heterologous systems or mouse dorsal root ganglion. Therefore, the human counterparts for these receptors, the MrgprX receptors and human H1R, remain poorly understood in native human sensory neurons where their functions are clinically important.”
- I think the discussion on pruritic priming could include more context about its potential clinical relevance.
We agree and have incorporated the reviewer’s suggestion into both the introduction and the discussion in several places to highlight the potential clinical significance of pruritic priming.
- 3, lines 11-13
“Several lines of evidence indicate that AEW-evoked scratching is a useful animal model of non-histaminergic itch and therefore may be a clinically relevant tool for the investigation of non-histaminergic pruritic diseases that respond poorly to antihistamines.”
- 5, lines 42-44
“…such latent sensitization, or “pruritic priming”, could be further investigated in future studies as a mechanism for the poorly understood but clinically important transition from acute to chronic pruritus.”
- 7, lines 12-14
“Clinically relevant pruritic diseases such as atopic dermatitis share mechanisms with peripheral pain, however, there are also important distinctions [47]. Thus, while pruriceptors can be sensitized and produce behavioral hypersensitivity in other models, the long-term latent sensitization underlying hyperalgesic priming was not matched with dry skin itch.”
- I suggest adding a dedicated section to explicitly discuss the strengths and limitations of the study.
We agree and note that Reviewer 3 also mentioned this suggestion in their critique. We have added such a section to the discussion as a dedicated paragraph on p. 7 lines 13-37. This paragraph has strengthened the manuscript and the suggestion is appreciated.
“A key strength of this study is that it is the first investigation of the novel concept of pruritic priming as a complement to the prior and ongoing elegant studies that have characterized hyperalgesic priming[18]. A rationale approach was taken to choose an appropriate model to test pruritic priming that mimicking key features of hyperalgesic priming, that is, recovery to baseline within several days and activation of non-peptidergic nociceptors and PKC signaling. Several limitations to the study should also be recognized. We only tested the AEW model using only a 5.5-day treatment protocol. It is possible a longer or more robust dry skin pruritus may yield a behavioral priming effect. To equate scratching magnitude with pain behavior magnitude is not straightforward, nevertheless we selected an itch model that resolved behaviorally in 4-6 days similar to the injection of carrageenan or nerve growth factor often used in hyperalgesic priming protocols. It should be noted that in addition to non-peptidergic fiber hyperinnervation, AEW also increased peptidergic innervation [28; 48] and that MrgrpA3 positive neurons express peptidergic markers as well[16], suggesting that the specific population of itch nociceptors might uniquely express both non-peptidergic and peptidergic markers making them a distinct subpopulation from the pure non-peptidergic nociceptors responsible for some types of hyperalgesic priming. Other pruritic dermatitis models such as the vitamin D3 analog [49] or imiquimod [50] application may yet yield pruritic priming. Additionally, we used only a single dose of pruritic compounds to test for priming near the midpoint of their dose-response, however it is possible lower doses may have revealed a priming effect. Finally, our study tested only male mice, and it remains an open question as to whether female mice could exhibit a different result.”
- I suggest testing varying doses of histamine, chloroquine, and β-alanine to examine whether a dose-dependent response exists, providing further insights into the pathways involved.
This would be an interesting additional experiment; however, we believe that dose-response experiments are beyond the scope of this manuscript. This issue is discussed now in the new paragraph on the limitations to the study (p. 7 lines 13-37). The doses chosen for each of the pruritogens used in this study are commonly used to examine scratching behavior in mice and are low enough to observe a potential increase in scratching, i.e., below the ceiling. Additionally, the results presented in this manuscript indicate a lack of pruritic priming for dry skin itch and no trend to suggest otherwise, therefore we believe adding additional doses would be unlikely to reveal qualitatively new information.
- The authors need to perform a simple RT-PCR analysis of MrgprA3 expression in skin samples from AEW-treated mice could validate the molecular findings at the tissue level.
We are not able to complete this experiment as it would require additional resources, reagents and equipment, and experimental protocols that are completely new to the study. To address the issue, on p. 6, line 19 we cite previous work from Zhu et al., 2017 (#33) that shows MrgprA3 is increased in the dorsal root ganglia in dry skin conditions and hyperinnervation of MrgprA3 in the epidermis during dry skin. These authors used an elegant transgenic approach to label MrgprA3 positive epidermal fibers with red fluorescent protein and quantified the innervation.
- I suggest adding a histological analysis at intermediate recovery time points (e.g., Days 3 and 4) to provide a more detailed view of the dynamics of epidermal repair.
It is true that this experiment would increase the resolution of the anatomical findings, however, it would not change the conclusions of the study. Other studies looking at AEW have shown epidermal thickening at early timepoints on the suggested days and we do not believe these data need to be replicated again. Our study was focused on the resolution of scratching behavior after cessation of AEW treatment, therefore we performed the anatomical analysis specifically on days relevant to the return to baseline of scratching.
Reviewer 2 Report
Comments and Suggestions for Authors
In this paper, the authors investigated whether a mechanism analogous to hyperalgesic priming could exist for pruritic using the AEW itchy skin mouse model. They conclude that a prior dry skin-induced pruritus is insufficient to produce a latent, primed state. However, I wonder whether 5.5 days of twice-per-day treatments of AEW is sufficient for latent pruritic priming if it exists. Have the authors checked for an AEW application longer than 5.5 days or repeated sets of the 5.5-day treatments of AEW?
Author Response
Review 2
In this paper, the authors investigated whether a mechanism analogous to hyperalgesic priming could exist for pruritic using the AEW itchy skin mouse model. They conclude that a prior dry skin-induced pruritus is insufficient to produce a latent, primed state. However, I wonder whether 5.5 days of twice-per-day treatments of AEW is sufficient for latent pruritic priming if it exists. Have the authors checked for an AEW application longer than 5.5 days or repeated sets of the 5.5-day treatments of AEW?
The reviewer raises an important question and a limitation of the study. We now address this issue in the discussion on p. 7 lines 22-27. The time course of AEW treatment was chosen to produce equivalent duration and intensity as could be reflected in literature on hyperalgesic priming. For hyperalgesic priming a single injection of NGF or carrageenan is sufficient to induce the priming effect. These stimuli evoke mild pain and hypersensitivity lasting approximately 3-4 days before becoming undetectable. In comparison, the chosen time course of AEW treatment matches well, with ongoing pruritus as evidenced by scratching behavior lasting several days before becoming undetectable after about 4 days from cessation of treatment. For hyperalgesic priming, multiple doses of NGF or carrageenan treatment are not required to induce the priming effect, and therefore we did not attempt multiple rounds or extended duration of AEW treatment. Nevertheless, we agree with the reviewer that it is possible that the effects of multiple or longer duration AEW treatment may yield latent sensitivity that was undetectable in this study. The following text has been added to the discussion to address this as a potential limitation on p. 7 lines 22-27:
“Several limitations to the study should also be recognized. We only tested the AEW model using only a 5.5-day treatment protocol. It is possible a longer or more robust dry skin pruritus may yield a behavioral priming effect. To equate scratching magnitude with pain behavior magnitude is not straightforward, nevertheless we selected an itch model that resolved behaviorally in 4-6 days similar to the injection of carrageenan or nerve growth factor often used in hyperalgesic priming protocols.”
Reviewer 3 Report
Comments and Suggestions for Authors
1) The Authors performed an interesting Brief Report entitled "Acetone-Ether-Water mouse model of persistent itch fully resolves without latent pruritic or cross-modality priming". It was established the time course for recovery from AEW-induced dry skin itch and then tested whether pruritogen-evoked scratching behavior could be primed, and tested whether cross-modality priming could be evoked, whereby dry skin itch or inflammatory pain could prime the other modality after a full recovery. In addition, the publication was approved by the University of Cincinnati Institutional Animal Care and Use Committee, and contains a statistical analysis of the data. However, I present some aspects that should be mentioned or improved in the manuscript.
2) Lines 66-67: Adult male C57Bl/6 mice (Jackson Labs) between 8-12 weeks were used for all experiments. Why didn't you use female animals? (the sex of the animals can influence the results…) How did you determine the sample size for the experiments? Is the sample size not small? Please comment on these aspects in the article.
3) Lines 98-100: After recovery, mice were injected intradermally with a 30-gauge insulin syringe into the previously treated cheek with a pruritogen: chloroquine (CQ, 100µg/10µl; Sigma), histamine (50µg/10µl; Sigma), or β-alanine (100mM; Sigma). Would it also be appropriate to test an H1 antihistamine drug? Please comment on these aspect in the article.
4) Figure 1, A: Day 7 has a considerable error bar (AEW). Is there any reason that can explain this fact?
5) Lines 192-193: Antihistamines are generally ineffective for dry skin itch... It should be mentioned that drugs are H1 antihistamines (for example, H2 antihistamines are used in the treatment of dyspepsia, peptic ulcers and gastroesophageal reflux disease).
6) The limitations of the study should be highlighted (without diminishing the value of the study).
Author Response
Review 3
The Authors performed an interesting Brief Report entitled "Acetone-Ether-Water mouse model of persistent itch fully resolves without latent pruritic or cross-modality priming". It was established the time course for recovery from AEW-induced dry skin itch and then tested whether pruritogen-evoked scratching behavior could be primed, and tested whether cross-modality priming could be evoked, whereby dry skin itch or inflammatory pain could prime the other modality after a full recovery. In addition, the publication was approved by the University of Cincinnati Institutional Animal Care and Use Committee, and contains a statistical analysis of the data. However, I present some aspects that should be mentioned or improved in the manuscript.
2) Lines 66-67: Adult male C57Bl/6 mice (Jackson Labs) between 8-12 weeks were used for all experiments. Why didn't you use female animals? (the sex of the animals can influence the results…) How did you determine the sample size for the experiments? Is the sample size not small? Please comment on these aspects in the article.
The reviewer is correct that female mice were not used in this study. We now address this issue explicitly in two separate places in the text:
- 3 lines 41-44
“Hyperalgesic priming is sexually dimorphic and female sex hormones protect against the development of priming rodents [22]. Therefore, to assess the first model of pruritic priming, only males were used in this study.”
- 7 lines 35-37
“Finally, our study tested only male mice, and it remains an open question as to whether female mice could exhibit a different result.”
Regarding sample size, the numbers used in this study are fairly standard in the literature and we have published several prior studies with similar numbers of animals using AEW dry skin and other models of pruritus (see for examples citations: #11, #36, #50, #51). Sample sizes were calculated to detect a difference of the mean 15% or greater between groups, which we consider to be biologically impactful. Based on our previous studies of itch and pruritus in mice and the expected variability in scratching behaviors across several types of pruritogens we have used in the past and in this study we assessed that 8 animals was sufficient.
3) Lines 98-100: After recovery, mice were injected intradermally with a 30-gauge insulin syringe into the previously treated cheek with a pruritogen: chloroquine (CQ, 100µg/10µl; Sigma), histamine (50µg/10µl; Sigma), or β-alanine (100mM; Sigma). Would it also be appropriate to test an H1 antihistamine drug? Please comment on these aspect in the article.
The use of an antihistamine in this study would be interesting if the results had been different. We hypothesized and tested that pruritic priming from dry skin would sensitize subsequent responses to histamine and increase scratching behavior. This enhanced itch would have been interpreted to indicate that pruritic nociceptors can be primed, and such priming could be an underlying mechanism for chronic pruritus. However, our results instead showed that dry skin produced no pruritic priming, and no enhanced response to histamine. Therefore, testing an H1 antihistamine would not any new information to the study.
4) Figure 1, A: Day 7 has a considerable error bar (AEW). Is there any reason that can explain this fact?
The reviewer correctly observes the highest level of variability occurs on day 7, which is the day of the highest level of scratching behavior. This behavior was measured as bouts of spontaneous scratching over a 60-minute period and some variability is inherent over this period; as more scratching occurs there is more variability inherent. Animals are free to behave (or even to sleep) during the measurement period which can add variability. Nevertheless, the bouts of scratching are in line with our previously published data on scratching behavior in mice (see: #11, #36, #50, #51) and the sample sizes are consistent with the literature and our previous work.
5) Lines 192-193: Antihistamines are generally ineffective for dry skin itch... It should be mentioned that drugs are H1 antihistamines (for example, H2 antihistamines are used in the treatment of dyspepsia, peptic ulcers and gastroesophageal reflux disease).
We thank the reviewer for this observation and have edited the manuscript on p. 6 line 24, p. 3 line 5 and line 13 to reflect that regarding itch and pruritus the antihistamines of note are H1R antihistamines.
6) The limitations of the study should be highlighted (without diminishing the value of the study).
The reviewer is correct to point this out and we have now added a new paragraph and several new citations to the discussion to better capture the limitations on p. 7 lines 13-37:
“A key strength of this study is that it is the first investigation of the novel concept of pruritic priming as a complement to the prior and ongoing elegant studies that have characterized hyperalgesic priming[18]. A rationale approach was taken to choose an appropriate model to test pruritic priming that mimicking key features of hyperalgesic priming, that is, recovery to baseline within several days and activation of non-peptidergic nociceptors and PKC signaling. Several limitations to the study should also be recognized. We only tested the AEW model using only a 5.5-day treatment protocol. It is possible a longer or more robust dry skin pruritus may yield a behavioral priming effect. To equate scratching magnitude with pain behavior magnitude is not straightforward, nevertheless we selected an itch model that resolved behaviorally in 4-6 days similar to the injection of carrageenan or nerve growth factor often used in hyperalgesic priming protocols. It should be noted that in addition to non-peptidergic fiber hyperinnervation, AEW also increased peptidergic innervation [28; 48] and that MrgrpA3 positive neurons express peptidergic markers as well[16], suggesting that the specific population of itch nociceptors might uniquely express both non-peptidergic and peptidergic markers making them a distinct subpopulation from the pure non-peptidergic nociceptors responsible for some types of hyperalgesic priming. Other pruritic dermatitis models such as the vitamin D3 analog [49] or imiquimod [50] application may yet yield pruritic priming. Additionally, we used only a single dose of pruritic compounds to test for priming near the midpoint of their dose-response, however it is possible lower doses may have revealed a priming effect. Finally, our study tested only male mice, and it remains an open question as to whether female mice could exhibit a different result.”
Round 2
Reviewer 1 Report
Comments and Suggestions for Authors
No more comments
Comments on the Quality of English LanguageThe English could be improved to more clearly express the research.
Reviewer 3 Report
Comments and Suggestions for Authors
Dear Authors,
With the changes performed,
The article has improved significantly and is now suitable for publication.